# Information Plane Analysis of Deep Neural Networks via Matrix–Based Rényi's Entropy and Tensor Kernels

## Abstract

Analyzing deep neural networks (DNNs) via information plane (IP) theory has gained tremendous attention recently as a tool to gain insight into, among others, their generalization ability. However, it is by no means obvious how to estimate mutual information (MI) between each hidden layer and the input/desired output, to construct the IP. For instance, hidden layers with many neurons require MI estimators with robustness towards the high dimensionality associated with such layers. MI estimators should also be able to naturally handle convolutional layers, while at the same time being computationally tractable to scale to large networks. None of the existing IP methods to date have been able to study truly deep Convolutional Neural Networks (CNNs), such as the e.g. VGG-16. In this paper, we propose an IP analysis using the new matrix–based Rényi's entropy coupled with tensor kernels over convolutional layers, leveraging the power of kernel methods to represent properties of the probability distribution independently of the dimensionality of the data. The obtained results shed new light on the previous literature concerning small-scale DNNs, however using a completely new approach. Importantly, the new framework enables us to provide the first comprehensive IP analysis of contemporary large-scale DNNs and CNNs, investigating the different training phases and providing new insights into the training dynamics of large-scale neural networks.

## 1 Introduction

Although Deep Neural Networks (DNNs) are at the core of most state–of–the art systems in computer vision, the theoretical understanding of such networks is still not at a satisfactory level (Shwartz-Ziv & Tishby, 2017). In order to provide insight into the inner workings of DNNs, the prospect of utilizing the Mutual Information (MI), a measure of dependency between two random variables, has recently garnered a significant amount of attention (Cheng et al., 2018; Noshad et al., 2019; Saxe et al., 2018; Shwartz-Ziv & Tishby, 2017; Yu et al., 2018; Yu & Principe, 2019). Given the input variable $X$ and the desired output $Y$ for a supervised learning task, a DNN is viewed as a transformation of $X$ into a representation that is favorable for obtaining a good prediction of $Y$. By treating the output of each hidden layer as a random variable $T$, one can model the MI $I(X;T)$ between $X$ and $T$. Likewise, the MI $I(T;Y)$ between $T$ and $Y$ can be modeled. The quantities $I(X;T)$ and $I(T;Y)$ span what is referred to as the Information Plane (IP). Several works have demonstrated that one may unveil interesting properties of the training dynamics by analyzing DNNs in the form of the IP (Yu & Principe, 2019; Goldfeld et al., 2019; Noshad et al., 2019; Chelombiev et al., 2019). Figure 1, produced using our proposed estimator, illustrates one such insight that is similar to the observations of Shwartz-Ziv & Tishby (2017), where training can be separated into two distinct phases, the fitting phase and the compression phase. This claim has been highly debated as subsequent research has linked the compression phase to saturation of neurons (Saxe et al., 2018) or clustering of the hidden representations (Goldfeld et al., 2019).

**Contributions** We propose a novel approach for estimating MI, wherein a kernel tensor-based estimator of Rényi's entropy allows us to provide the first analysis of large-scale DNNs as commonly found in state-of-the-art methods. We further highlight that the multivariate matrix–based

approach, proposed by Yu et al. (2019), can be viewed as a special case of our approach. However, our proposed method alleviates numerical instabilities associated with the multivariate matrix–based approach, which enables estimation of entropy for high-dimensional multivariate data. Further, using the proposed estimator, we investigate the claim of Cheng et al. (2018) that the entropy $H(X) \approx I(T; X)$ and $H(Y) \approx I(T; Y)$ in high dimensions (in which case MI-based analysis would be meaningless) and illustrate that this does not hold for our estimator. Finally, our results indicate that the compression phase is apparent mostly for the training data, particularly for more challenging datasets. By utilizing a technique such as early-stopping, a common technique to avoid overfitting, training tends to stop before the compression phase occurs (see Figure 1). This may indicate that the compression phase is linked to the overfitting phenomena.

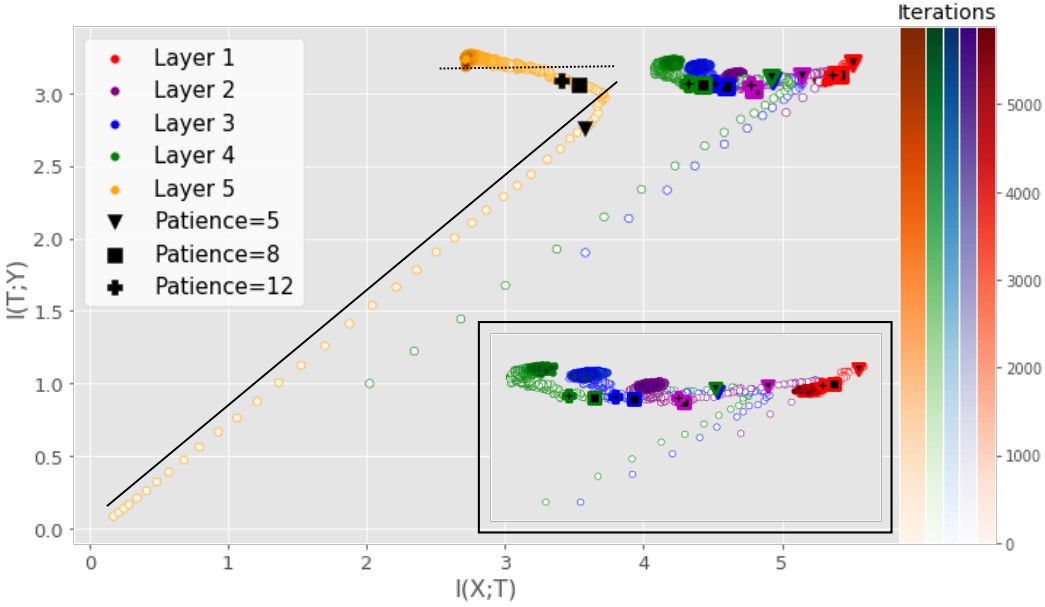

Figure 1: IP obtained using our proposed estimator for a small DNN averaged over 5 training runs. The solid black line illustrates the fitting phase while the dotted black line illustrates the compression phase. The iterations at which early stopping would be performed assuming a given patience parameter are highlighted. Here, patience denotes the number of iterations that need to pass without progress on a validation set before training is stopped to avoid overfitting. It can be observed that for low patience values, training will stop before the compression phase. For the benefit of the reader, the bottom right corner displays a magnified version of the first four layers.

## 2 RELATED WORK

Analyzing DNNs in the IP was first proposed by Tishby & Zaslavsky (2015) and later demonstrated by Shwartz-Ziv & Tishby (2017). Among other results, the authors studied the evolution of the IP during the training process of DNNs and noted that the process was composed of two different phases. First, an initial fitting phase where $I(T; Y)$ increases, followed by a phase of compression where $I(X; T)$ decreases. These results were later questioned by Saxe et al. (2018), who argued that the compression phase is not a general property of the DNN training process, but rather an effect of different activation functions. However, a recent study by Noshad et al. (2019) seems to support the claim of a compression phase, regardless of the activation function. The authors argue that the base estimator of MI utilized in Saxe et al. (2018) might not be accurate enough and demonstrate that a compression phase does occur, but the amount of compression can vary between different activation functions. Another recent study by (Chelombiev et al., 2019) also reported a compression phase, but highlighted the importance of adaptive MI estimators. They also showed that when L2-regularization was included in the training compression was observed, regardless of activation function. Also, some recent studies have discussed the limitations of the IP framework for analysis and optimization for particular types of DNN (Kolchinsky et al. (2019); Amjad & Geiger (2019)).

On a different note, Cheng et al. (2018) proposed an evaluation framework for DNNs based on the IP and demonstrated that MI can be used to infer the capability of DNNs to recognize objects for an image classification task. Furthermore, the authors argue that when the number of neurons in a hidden layer grows large, $I(T;X)$ and $I(Y;T)$ barely change and are, using Cheng et al. (2018) terminology, approximately deterministic, i.e. $I(T;X) \approx H(X)$ and $I(T;Y) \approx H(Y)$. Therefore, they only model the MI between $X$ and the last hidden layer, that is the output of the network, and the last hidden layer and $Y$.

Yu & Principe (2019) initially proposed to utilize empirical estimators for Rényi's MI for investigating different data processing inequalities in stacked autoencoders (SAEs). They also claimed that the compression phase in the IP of SAEs is determined by the values of the SAE bottleneck layer size and the intrinsic dimensionality of the given data. Yu et al. (2019) also extended the empirical estimators for Rényi's MI to the multivariate scenario and applied the new estimator to simple but realistic Convolutional Neural Networks (CNNs) (Yu et al., 2018). However, the results so far suffer from high computational burden and are hard to generalize to deep and large-scale CNNs.

## 3 MATRIX–BASED MUTUAL INFORMATION

Here, we review Rényi's $\alpha$-order entropy and its multivariate extension proposed by Yu et al. (2019).

### 3.1 MATRIX–BASED RÉNYI'S ALPHA-ORDER ENTROPY

Rényi's $\alpha$-order entropy is a generalization of Shannon's entropy (Shannon, 1948; Renyi, 1961). For a random variable $X$ with probability density function (PDF) $f(\mathbf{x})$ with support $\mathcal{X}$, Rényi's $\alpha$-order entropy is defined as:

$$H_\alpha(f) = \frac{1}{1-\alpha} \log \int_\mathcal{X} f^\alpha(\mathbf{x}) d\mathbf{x} \tag{1}$$

Equation 1 has been widely applied in machine learning (Principe, 2010), and the particular case of $\alpha = 2$, combined with Parzen window density estimation (Parzen, 1962), form the basis for Information Theoretic Learning (Principe, 2010). However, accurately estimating PDFs in high-dimensional data, which is typically the case for DNNs, is a challenging task.

To avoid the problem of high-dimensional PDF estimation, Giraldo et al. (2012) proposed a non-parametric framework for estimating entropy directly from data without resorting to kernel density estimation:

**Definition 3.1.** (Giraldo et al., 2012) Let $\mathbf{x}_i \in \mathcal{X}$, $i = 1, 2, \ldots, N$ denote data points and let $\kappa : \mathcal{X} \times \mathcal{X} \mapsto \mathbb{R}$ be an infinitely divisible positive definite kernel (Bhatia, 2006). Given the kernel matrix $\mathbf{K} \in \mathbb{R}^{N \times N}$ with elements $(\mathbf{K})_{ij} = \kappa(\mathbf{x}_i, \mathbf{x}_j)$ and the matrix $\mathbf{A}$, $(\mathbf{A})_{ij} = \frac{1}{N} \frac{(\mathbf{K})_{ij}}{\sqrt{(\mathbf{K})_{ii}(\mathbf{K})_{jj}}}$, the matrix–based Rényi's $\alpha$–order entropy is given by

$$S_\alpha(\mathbf{A}) = \frac{1}{1-\alpha} \log_2 \left( \text{tr}(\mathbf{A}^\alpha) \right) = \frac{1}{1-\alpha} \log_2 \left[ \sum_{i=1}^N \lambda_i(\mathbf{A})^\alpha \right]. \tag{2}$$

Here, $\lambda_i(\mathbf{A})$ denotes the i[th] eigenvalue of the matrix $\mathbf{A}$. The properties of this quantity was analysed in detail in Giraldo et al. (2012). It was shown that the kernel matrix $\mathbf{A}$, obtained from the raw data, acts much as a density matrix in quantum information theory (Nielsen & Chuang, 2011). It was further shown that the (Gram) matrix $\mathbf{A}$ is related to an empirical covariance operator on embeddings of probability distributions in a Reproducing Kernel Hilbert Space (RKHS). This is similar to the approach of maximum mean discrepancy and the kernel mean embedding (Gretton et al., 2012; Muandet et al., 2017). Moreover, Giraldo et al. (2012) showed that under certain conditions Equation 2 converges to the trace of the underlying covariance operator, as shown in Proposition B.1 in Appendix B. Notice that the dimensionality of the data does not appear in Proposition B.1. This means that $S_\alpha(\mathbf{A})$ captures properties of the distribution, with a certain robustness with respect to high-dimensional data. This is a beneficial property compared to KNN and KDE based information estimators used in previous works (Saxe et al., 2018; Chelombiev et al., 2019), which have difficulties handling high-dimensional data (Kwak & Chong-Ho Choi, 2002). Not all estimators of entropy have the same property (Paninski, 2003). Certain approaches developed for estimating the Shannon

entropy suffer from the curse of dimensionality (Kwak & Chong-Ho Choi, 2002). Also, there is no need for any binning procedure utilized in previous works (Shwartz-Ziv & Tishby, 2017), which are known to struggle with the relu activation function commonly used in DNN (Saxe et al., 2018). In Appendix A we have conducted experiments on synthetic data to illustrate the behaviours of these estimators for high-dimensional data.

**Remark 1.** In the limit when $\alpha \to 1$, Equation 2 reduces to the matrix–based von Neumann entropy (Nielsen & Chuang, 2011) that resembles Shannon's definition over probability states, and can be expressed as

$$\lim_{\alpha \to 1} S_\alpha(\mathbf{A}) = -\sum_{i=1}^{N} \lambda_i(\mathbf{A}) \log_2[\lambda_i(\mathbf{A})]. \tag{3}$$

For completeness, the proof of Equation 3 can be found in Appendix C.

In addition to the definition of matrix based entropy, Giraldo et al. (2012) define the joint entropy between $\mathbf{x} \in \mathcal{X}$ and $\mathbf{y} \in \mathcal{Y}$ as

$$S_\alpha(\mathbf{A}_\mathcal{X}, \mathbf{A}_\mathcal{Y}) = S_\alpha\left(\frac{\mathbf{A}_\mathcal{X} \circ \mathbf{A}_\mathcal{Y}}{\text{tr}(\mathbf{A}_\mathcal{X} \circ \mathbf{A}_\mathcal{Y})}\right), \tag{4}$$

where $\mathbf{x}_i$ and $\mathbf{y}_i$ are two different representations of the same object and $\circ$ denotes the Hadamard product. Finally, the MI is, similar to Shannon's formulation, defined as

$$I_\alpha(\mathbf{A}_\mathcal{X}; \mathbf{A}_\mathcal{Y}) = S_\alpha(\mathbf{A}_\mathcal{X}) + S_\alpha(\mathbf{A}_\mathcal{Y}) - S_\alpha(\mathbf{A}_\mathcal{X}, \mathbf{A}_\mathcal{Y}). \tag{5}$$

### 3.2 MULTIVARIATE MATRIX-BASED RÉNYI'S ALPHA-ENTROPY FUNCTIONALS

The matrix-based Rényi's $\alpha$-order entropy functional is not suitable for estimating the amount of information of the features produced by a convolutional layer in a DNN as the output consists of $C$ feature maps, each represented by their own matrix, that characterize different properties of the same sample. Yu et al. (2019) proposed a multivariate extension of the matrix–based Rényi's $\alpha$-order entropy, which computes the joint-entropy among $C$ variables as

$$S_\alpha(\mathbf{A}_1, \ldots, \mathbf{A}_C) = S_\alpha\left(\frac{\mathbf{A}_1 \circ \ldots \circ \mathbf{A}_C}{\text{tr}(\mathbf{A}_1 \circ \ldots \circ \mathbf{A}_C)}\right), \tag{6}$$

where $(\mathbf{A}_1)_{ij} = \kappa_1(\mathbf{x}_i^{(1)}, \mathbf{x}_j^{(1)})$, ..., $(\mathbf{A}_C)_{ij} = \kappa_C(\mathbf{x}_i^{(C)}, \mathbf{x}_j^{(C)})$. Yu et al. (2018) also demonstrated how Equation 6 could be utilized for analyzing synergy and redundancy of convolutional layers in DNN, but noted that this formulation can encounter difficulties when the number of feature maps increases, such as in more complex CNNs. Difficulties arise due to the Hadamard products in Equation 6, given that each element of $A_c$, $c \in \{1, 2, \ldots, C\}$, takes on a value between 0 and $\frac{1}{N}$, and the product of $C$ such elements thus tends towards 0 as $C$ grows. Yu et al. (2018) reported such challenges when attempting to model the IP of the VGG16 (Simonyan & Zisserman, 2015). We have illustrated how this numerical instability manifests itself in Appendix E.

## 4 TENSOR-BASED MUTUAL INFORMATION

To invoke information theoretic quantities of features produced by convolutional layers and address the limitations discussed above, we introduce our tensor-based approach for utilizing entropy and MI in DNNs, and show that the multivariate approach in section 3.2 arises as a special case.

### 4.1 TENSOR KERNELS FOR MUTUAL INFORMATION ESTIMATION

The output of a convolutional layer is represented as a tensor $\mathbb{X}_i \in \mathbb{R}^C \otimes \mathbb{R}^H \otimes \mathbb{R}^W$ for a data point $i$. As discussed above, the matrix–based Rényi's $\alpha$-entropy can not include tensor data without modifications. To handle the tensor based nature of convolutional layers we propose to utilize tensor kernels (Signoretto et al., 2011) to produce a kernel matrix for the output of a convolutional layer. A tensor formulation of the radial basis function (RBF) kernel can be stated as

$$\kappa_{\text{ten}}(\mathbb{X}_i, \mathbb{X}_j) = e^{-\frac{1}{\sigma^2} \|\mathbb{X}_i - \mathbb{X}_j\|_F^2}, \tag{7}$$

where $\| \cdot \|_F$ denotes the Hilbert-Frobenius norm (Signoretto et al., 2011) and $\sigma$ is the kernel width parameter. In practice, the tensor kernel in Equation 7 can be computed by reshaping the tensor into a vectorized representation while replacing the Hilbert-Frobenius norm with a Euclidean norm. We compute the MI in Equation 5 by replacing the matrix $\mathbf{A}$ with

$$
\begin{aligned}
(\mathbf{A}_{\text{ten}})_{ij} &= \frac{1}{N} \frac{(\mathbf{K}_{\text{ten}})_{ij}}{\sqrt{(\mathbf{K}_{\text{ten}})_{ii}(\mathbf{K}_{\text{ten}})_{jj}}} \\
&= \frac{1}{N} \kappa_{\text{ten}}(\mathbb{X}_i, \mathbb{X}_j).
\end{aligned}
\tag{8}
$$

While Equation 7 provides the simplest and most intuitive approach for using kernels with tensor data, it does have its limitations. Namely, a tensor kernel that simply vectorizes the tensor ignores the inter-component structures within and between the respective tensor (Signoretto et al., 2011). For simple tensor data such structures might not be present and a tensor kernel as described above can suffice, however, other tensor kernels do exist, such as for instance the matricization-based tensor kernels Signoretto et al. (2011). In this work we have chosen the tensor kernel defined in Equation 7 for its simplicity and computational benefits, which come from the fact that the entropy and joint entropy are computed batch-wise by finding the eigenvalues of a kernel matrix, or the eigenvalues of the Hadamard product of two kernel matrices, and utilizing Equation 2. Nevertheless, exploring structure preserving kernels can be an interesting research path in future works. In Appendix E we have included a simple example towards this direction, where the tensor kernels described in this paper is compared to a matricization-based tensor kernel. Note that the multivariate approach described in Section 3.2 can be regarded as a special case of our proposed method given under certain assumptions and a proof is provided in Appendix D.

## 4.2 CHOOSING THE KERNEL WIDTH

With methods involving RBF kernels, the choice of the kernel width parameter, $\sigma$, is always critical. For supervised learning problems, one might choose this parameter by cross-validation based on validation accuracy, while in unsupervised problems one might use a rule of thumb (Shi & Malik, 2000; Shi et al., 2009; Silverman, 1986). However, in the case of estimating MI in DNNs, the data is often high dimensional, in which case unsupervised rules of thumb often fail (Shi et al., 2009).

In this work, we choose $\sigma$ based on an optimality criterion. Intuitively, one can make the following observation: A good kernel matrix should reveal the class structures present in the data. This can be accomplished by maximizing the so–called *kernel alignment* loss (Cristianini et al., 2002) between the kernel matrix of a given layer, $\mathbf{K}_\sigma$, and the label kernel matrix, $\mathbf{K}_y$. The kernel alignment loss is defined as

$$
A(\mathbf{K}_a, \mathbf{K}_b) = \frac{\langle \mathbf{K}_a, \mathbf{K}_b \rangle_F}{\|\mathbf{K}_a\|_F \|\mathbf{K}_b\|_F},
\tag{9}
$$

where $\|\cdot\|_F$ and $\langle \cdot, \cdot \rangle_F$ denotes the Frobenius norm and inner product, respectively. Thus, we choose our optimal $\sigma$ as

$$
\sigma^* = \arg\max_\sigma A(\mathbf{K}_\sigma, \mathbf{K}_y).
$$

To stabilize the $\sigma$ values across mini batches, we employ an exponential moving average, such that in layer $\ell$ at iteration $t$, we have

$$
\sigma_{\ell,t} = \beta \sigma_{\ell,t-1} + (1 - \beta)\sigma^*_{\ell,t},
$$

where $\beta \in [0, 1]$ and $\sigma_{\ell,1} = \sigma^*_{\ell,1}$.

## 5 EXPERIMENTAL RESULTS

We evaluate our approach by comparing it to previous results obtained on small networks by considering the MNIST dataset and an Multi Layer Perceptron (MLP) architecture that was inspired by Saxe et al. (2018). We further compare to a small CNN architecture similar to that of Noshad et al. (2019), before considering large networks, namely VGG16, and a more challenging dataset, namely CIFAR-10. Note, unless stated otherwise, we use CNN to denote the small CNN architecture. Details about the MLP and the CNN utilized in these experiments can be found in Appendix F. All MI estimates were computed using Equation 3, 4 and 5 and the tensor approach described in Section 4.

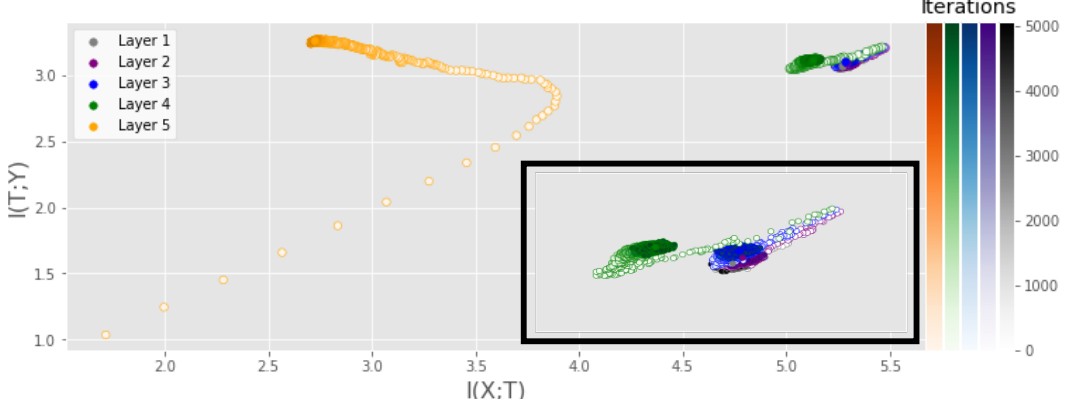

Figure 2: IP of a CNN consisting of three convolutional layers with 4, 8 and 12 filters and one fully connected layer with 256 neurons and a ReLU activation function in in each hidden layer. MI was computed using the training data of the MNIST dataset and averaged over 5 runs.

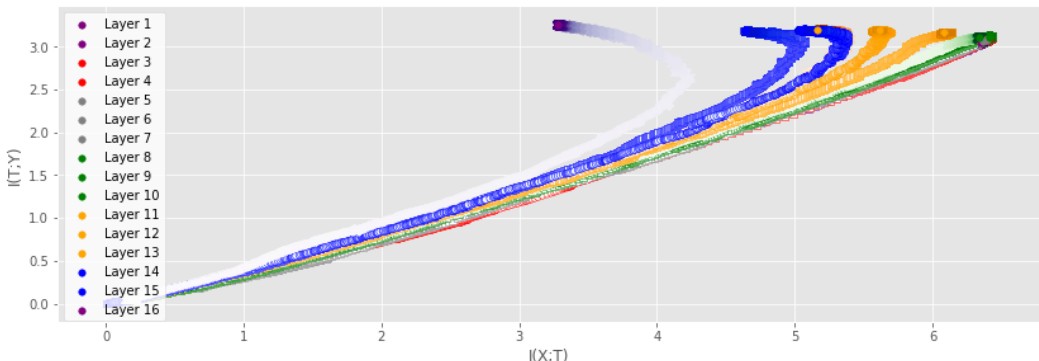

Figure 3: IP of the VGG16 on the CIFAR-10 dataset. MI was estimated using the training data and averaged over 2 runs. Color saturation increases as training progresses. Both the fitting phase and the compression phase is clearly visible for several layers.

Since the matrix-based Rényi MI is computed at batch level a certain degree of noise is present. We employ a moving average smoothing approach where each sample is averaged over $k$ mini-batches. For the MLP and CNN experiments we use $k = 10$ and for the VGG16 we use $k = 50$. We use a batch size of 100 samples, and determine the kernel width using the kernel alignment loss defined in Equation 9. For each hidden layer, we chose the kernel width that maximizes the kernel alignment loss in the range 0.1 and 10 times the mean distance between the samples in one mini-batch. Initially, we sample 75 equally spaced values for the kernel width in the given range for the MLP and CNN and 300 values for the VGG16 network. During training, we dynamically reduce the number of samples to 50 and 100 respectively in to reduce computational complexity and motivated by the fact that the kernel width remains relatively stable during the latter part of training (illustrated in Appendix I). We chose the range 0.1 and 10 times the mean distance between the samples in one mini-batch to avoid the kernel width becoming too small and to ensure that we cover a wide enough range of possible values. For the input kernel width we empirically evaluated values in the range 2-16 and found consistent results for values in the range 4-12. All our experiments were conducted with an input kernel width of 8. For the label kernel matrix, we want a kernel width that is as small as possible to approach an ideal kernel matrix, but also avoid numerical instabilities. For all our experiments we use a value of 0.1 for the kernel width of the label kernel matrix.

**Comparison to Previous Approaches**   First, we study the IP of the MLP examined in previous works on DNN analysis using information theory (Noshad et al., 2019; Saxe et al., 2018). We utilize stochastic gradient descent with a learning rate of 0.09, a cross-entropy loss function, and repeat the experiment 5 times. Figure 1 displays the IP of the MLP with a ReLU activation function in each hidden layer. MI was estimated using the training data of the MNIST dataset. A similar experiment

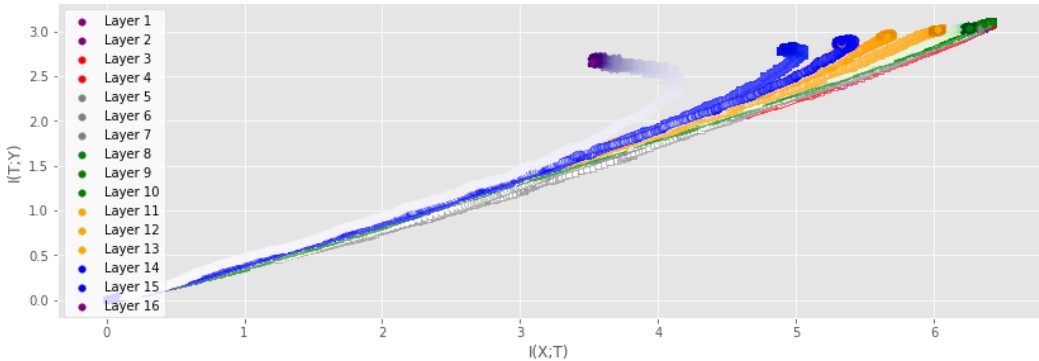

Figure 4: IP of the VGG16 on the CIFAR-10 dataset. MI was estimated using the test data and averaged over 2 runs. Color saturation increases as training progresses. The fitting phase is clearly visible while the compression phase can only be seen in the output layer.

was performed with the tanh activation function, obtaining similar results. The interested reader can find these results in Appendix G.

From Figure 1 one can clearly observe a fitting phase, where both $I(T;X)$ and $I(Y;T)$ increases rapidly, followed by a compression phase where $I(T;X)$ decrease and $I(Y;T)$ remains unchanged. Also note that $I(Y;T)$ for the output layer (layer 5 in Figure 1) stabilizes at an approximate value of $\log_2(10)$, which is to be expected. This can be seen by noting that when the network achieves approximately $100\%$ accuracy, $I(Y;\hat{Y}) \approx S(Y)$, where $\hat{Y}$ denotes the output of the network, since $Y$ and $\hat{Y}$ will be approximately identical and the MI between a variable and itself is just the entropy of the variable. The entropy of $Y$ is estimated using Equation 3, which requires the computation of the eigenvalues of the label kernel matrix $\frac{1}{N}\mathbf{K}_y$. For the ideal case, where $(\mathbf{K}_y)_{ij} = 1$ if $y_i = y_j$ and zero otherwise, $\mathbf{K}_y$ is a rank $K$ matrix, where $K$ is the number of classes in the data. Thus, $\frac{1}{N}\mathbf{K}_y$ has $K$ non–zero eigenvalues which are given by $\lambda_k(\frac{1}{N}\mathbf{K}_y) = \frac{1}{N}\lambda_k(\mathbf{K}_y) = \frac{N_{c_k}}{N}$, where $N_{c_k}$ is the number of datapoints in class $k$, $k = 1, 2, \ldots, K$. Furthermore, if the dataset is balanced we have $N_{c_1} = N_{c_2} = \ldots = N_{c_K} \equiv N_c$. Then, $\lambda_k\left(\frac{1}{N}\mathbf{K}_y\right) = \frac{N_c}{N} = \frac{1}{K}$, which gives us the entropy estimate

$$
\begin{aligned}
S\left(\frac{1}{N}\mathbf{K}_y\right) &= -\sum_{k=1}^{K} \lambda_k\left(\frac{1}{N}\mathbf{K}_y\right) \log_2\left[\lambda_k\left(\frac{1}{N}\mathbf{K}_y\right)\right] \\
&= -\sum_{k=1}^{K} \frac{1}{K} \log_2\left[\frac{1}{K}\right] = \log_2[K].
\end{aligned}
\tag{10}
$$

Next we examine the IP of a CNN, similar to that studied by Noshad et al. (2019), with a similar experimental setup as for the MLP experiment. Figure 2 displays the IP of the CNN with a ReLU activation function in all hidden layers. A similar experiment was conducted using the tanh activation function and can be found in Appendix H. While the output layer behaves similarly to that of the MLP, the preceding layers show much less movement. In particular, no fitting phase is observed, which might be a result of the convolutional layers being able to extract the necessary information in very few iterations. Note that the output layer is again settling at the expected value $\log_2(10)$, similar to the MLP, as it also achieves close to $100\%$ accuracy.

**Increasing DNN size** Finally, we analyze the IP of the VGG16 network on the CIFAR-10 dataset, with the same experimental setup as in the previous experiments. To our knowledge, this is the first time that the full IP has been modeled for such a large-scale network. Figure 3 and 4 show the IP when computing the MI for the training dataset and the test dataset respectively. For the training dataset, we can clearly observe the same trend as for the smaller networks, where layers experience a fitting phase during the early stages of training and a compression phase in the later stage. Note, that the compression phase is less prominent for the testing dataset. Also note the difference between the final values of $I(Y;T)$ for the output layer estimated using the training and test data, which is a result of the different accuracy achieved on the training data ($\approx 100\%$) and test data ($\approx 90\%$). Cheng et al. (2018) claim that $I(T;X) \approx H(X)$ and $I(Y;T) \approx H(Y)$ for high dimensional data,

and highlight particular difficulties with computing the MI between convolutional layers and the input/output. However, this statement is dependent on their particular estimator for the MI, and the results presented in Figure 3 and 4 demonstrate that neither $I(T; X)$ nor $I(Y; T)$ is deterministic for our proposed quantity.

**Effect of Early Stopping**  We also investigate the effect of using early stopping on the IP described above. Early stopping is a regularization technique where the validation accuracy is monitored and training is stopped if the validation accuracy does not increase for a set number of iteration, often referred to as the patience hyperparameter. Figure 1 displays the results of monitoring where the training would stop if the early stopping procedure was applied for different values of patience. For a patience of 5 iterations the network training would stop before the compression phase takes place for several of the layers. For larger patience values, the effects of the compression phase can be observed before training is stopped. Early stopping is a procedure intended to prevent the network from overfitting, which may imply that the compression phase observed in the IP of DNNs can be related to overfitting. The interested reader can find further experiments on the compression phase and early stopping in Appendix J.

**Data Processing Inequality**  A DNN consists of a chain of mappings from the input, through the hidden layers and to the output. One can interpret a DNN as a Markov chain (Shwartz-Ziv & Tishby, 2017; Yu & Principe, 2019) that defines an information path (Shwartz-Ziv & Tishby, 2017), which should satisfy the following Data Processing Inequality (Cover & Thomas, 2006):

$$I(X; T_1) \geq I(X; T_2) \geq \ldots \geq I(X; T_L), \tag{11}$$

where $L$ is the number of layers in the network. An indication of a reasonable measure of MI is that it should uphold the DPI. Figure 11 in Appendix K illustrates the mean difference in MI between two subsequent layers in the MLP and VGG16 network. Positive numbers indicate that MI decreases, thus indicating compliance with the DPI. We observe that our quantity complies with the DPI for all layers in the MLP and all except one in the VGG16 network.

## 6  CONCLUSION

In this work, we propose a novel framework for analyzing DNNs from a MI perspective using a tensor-based estimate of the Rényi's $\alpha$-order entropy. Our experiments illustrate that the proposed approach scales to large DNNs, which allows us to provide insights into the training dynamics. We observe that the compression phase in neural network training tends to be more prominent when MI is estimated on the training set and that commonly used early-stopping criteria tend to stop training before or at the onset of the compression phase. This could imply that the compression phase is linked to overfitting. Furthermore, we showed that, for our tensor-based approach, the claim that $H(X) \approx I(T; X)$ and $H(Y) \approx I(T; Y)$ does not hold. We believe that our proposed approach can provide new insight and facilitate a more theoretical understanding of DNNs.

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

# A    VALIDATION OF MATRIX–BASED ESTIMATORS ON HIGH-DIMENSIONAL SYNTHETIC DATA

To examine the behaviour of the matrix–based estimators described in Section 3 we have conducted a simple experiment on estimating entropy and mutual information in 100-dimensional data following a normal distribution. First, we generate 500 samples from six 100-dimensional normal distribution with 0 mean and variance $\{0.25, 0.20, 0.15, 0.1, 0.05, 0.01\}$ and estimate the entropy of the resulting distributions. The results of this experiment is displayed in Figure 5, where the leftmost figure displays a two dimensional representation of the data and the the rightmost figure shows the estimated entropy for different values of the variance. In the rightmost plot, entropy is estimated using all samples and in a batch-wise setting, with batches of size 100 (like in the main paper). The experiment shows how the estimated entropy decreases as the variance of the distribution decreases, as expected. Also, it shows how the batch-wise approximation produce similar results as the estimates using the full dataset. Second, we generate 500 samples from a 100-dimensional normal distributions with mean 0 and variance 0.25, and 500 samples from six 100-dimensional normal distribution with mean 1 and variance $\{0.25, 0.20, 0.15, 0.1, 0.05, 0.01\}$ and estimate the MI between the distributions. In the rightmost plot, MI is estimated using all samples and in a batch-wise setting, with batches of size 100 (like in the main paper). The results of this experiment is depicted in Figure 6, where the leftmost figure displays a two dimensional representation of the data and rightmost figure shows the estimated MI as variance of one of the distributions is decreased and less of the distributions overlap. As the variance of the second distribution decreases and the distributions overlap less and less, the MI estimates also decrease, demonstrating that the estimators are able to capture dependencies in a high-dimensional setting. Moreover, the batch-wise approximation produces similar estimates as the full dataset approach.

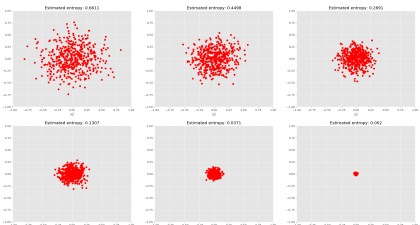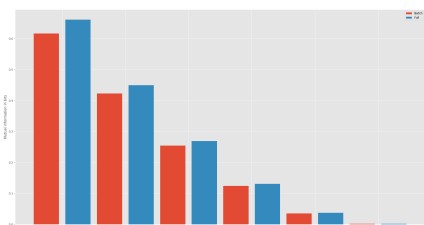

Figure 5: Leftmost figure displays a 2-dimensional illustration of the data described in Appendix A and rightmost figure shows how the entropy decreases as the variance of the distribution decreases. Entropy is estimated using Equation 2 on a 100-dimensional normal distribution, both with all samples (displayed in blue) and with batches of size 100 (displayed in red).

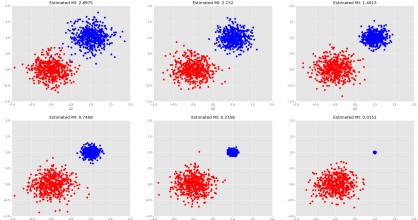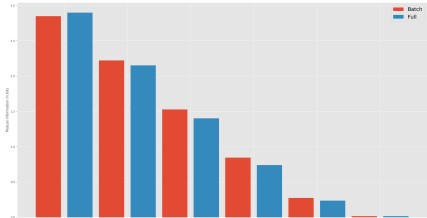

Figure 6: Leftmost figure displays a 2-dimensional illustration of the data described in Appendix A and rightmost figure shows how mutual information decreases as less of the distributions overlap. Mutual information is estimated using Equation 5 on two 100-dimensional normal distribution, using all samples (displayed in blue) and with batches of size 100 (displayed in red).

# B  BOUND ON MATRIX–BASED ENTROPY WITH RESPECT TO COVARIANCE OPERATOR

The properties of Equation 2 was analysed in detail in Giraldo et al. (2012). It was shown that the kernel matrix A, obtained from the raw data, acts much as a density matrix similar to quantum information theory (Nielsen & Chuang, 2011). It was further shown that the kernel matrix is related to an empirical covariance operator on embeddings of probability distributions in a RKHS. This is similar to the approach of maximum mean discrepancy and the kernel mean embedding (Gretton et al., 2012; Muandet et al., 2017). The connection with the data population can be shown via the theory of covariance operators. The covariance operator $G : \mathcal{H} \to \mathcal{H}$ is defined through the bilinear form

$$\mathcal{G}(f,g) = \langle f, Gg \rangle = \int_{\mathcal{X}} \langle f, \psi(\mathbf{x}) \rangle \langle \psi(\mathbf{x}), g \rangle \, d\mathbb{P}_{\mathcal{X}}(\mathbf{x}) = E_{\mathcal{X}} \{ f(X)g(Y) \}$$

where $\mathbb{P}_{\mathcal{X}}$ is a probability measure and $f, g \in \mathcal{H}$. Based on the empirical distribution $\mathbb{P}_N = \frac{1}{N} \sum_{i=1}^{N} \delta_{\mathbf{x}_i}(\mathbf{x})$, the empirical version $\hat{G}$ of $G$ obtained from a sample $\mathbf{x}_i$ of size N is given by:

$$\left\langle f, \hat{G}_N g \right\rangle = \hat{\mathcal{G}}(f,g) = \int_{\mathcal{X}} \langle f, \psi(\mathbf{x}) \rangle \langle \psi(\mathbf{x}), g \rangle \, d\mathbb{P}_{\mathcal{X}}(\mathbf{x}) = \frac{1}{N} \sum_{i=1}^{N} \langle f, \psi(\mathbf{x}_i) \rangle \langle \psi(\mathbf{x}_i), g \rangle$$

By analyzing the spectrum of $\hat{G}$ and $G$, Giraldo et al. (2012) showed the the difference between $\mathrm{tr}(G)$ and $\mathrm{tr}(\hat{G})$ can be bounded, as stated in the following proposition:

**Proposition B.1.** *Let $\mathbb{P}_N = \frac{1}{N} \sum_{i=1}^{N} \delta_{\mathbf{x}_i}(\mathbf{x})$ be the empirical distribution. Then, as a consequence of Proposition 6.1 in Giraldo et al. (2012), $\mathrm{tr}\left[ \hat{G}_N^\alpha \right] = \mathrm{tr}\left[ \left( \frac{1}{N}\mathbf{K} \right)^\alpha \right]$. The difference between $\mathrm{tr}(G)$ and $\mathrm{tr}(\hat{G})$ can be bounded under the conditions of Theorem 6.2 in Giraldo et al. (2012) and for $\alpha > 1$, with probability 1-δ*

$$\left| \mathrm{tr}\left( G^\alpha \right) - \mathrm{tr}\left( \hat{G}_N^\alpha \right) \right| \leq \alpha C \sqrt{\frac{2 \log \frac{2}{\delta}}{N}} \tag{12}$$

*where C is a compact self-adjoint operator.*

# C  PROOF OF EQUATION 3 IN SECTION 3

*Proof.*

$$\lim_{\alpha \to 1} S_\alpha(\mathbf{A}) = \lim_{\alpha \to 1} \frac{1}{1-\alpha} \log_2 \left( \sum_{i=1}^{n} \lambda_i^\alpha \right) \to \frac{0}{0},$$

since $\sum_{i=1}^{N} \lambda_i = \mathrm{tr}(\mathbf{A}) = 1$. L'Hôpital's rule yields

$$
\begin{aligned}
\lim_{\alpha \to 1} S_\alpha(\mathbf{A}) &= \lim_{\alpha \to 1} \frac{\frac{\partial}{\partial \alpha} \log_2 \left[ \sum_{i=1}^{n} \lambda_i(\mathbf{A})^\alpha \right]}{\frac{\partial}{\partial \alpha}(1-\alpha)} \\
&= -\frac{1}{\ln 2} \lim_{\alpha \to 1} \frac{\sum_{i=1}^{n} \lambda_i(\mathbf{A})^\alpha \ln[\lambda_i(\mathbf{A})]}{|\sum_{i=1}^{n} \lambda_i(\mathbf{A})^\alpha|} \\
&= -\sum_{i=1}^{n} \lambda_i(\mathbf{A}) \log_2[\lambda_i(\mathbf{A})].
\end{aligned}
\tag{13}
$$

□

## D    Tensor-Based Approach Contains Multivariate Approach as Special Case

Let $\mathbf{x}_i(\ell) \in \mathbb{R}^{HWC}$ denote the vector representation of data point $i$ in layer $\ell$ and let $\mathbf{x}_i^{(c)}(\ell) \in \mathbb{R}^{HW}$ denote its representation produced by filter $c$. In the following, we omit the layer index for ease of notation, but assume it is fixed. Consider the case when $\kappa_c(\cdot, \cdot)$ is an RBF kernel with kernel width parameter $\sigma_c$. That is $\kappa_c(\mathbf{x}_i^{(c)}, \mathbf{x}_j^{(c)}) = e^{-\frac{1}{\sigma_c^2}\|\mathbf{x}_i^{(c)}-\mathbf{x}_j^{(c)}\|^2}$. In this case, $\mathbf{A}_c = \frac{1}{N}\mathbf{K}_c$ and

$$
\begin{aligned}
\frac{\mathbf{A}_1 \circ \ldots \circ \mathbf{A}_C}{\operatorname{tr}(\mathbf{A}_1 \circ \ldots \circ \mathbf{A}_C)} &= \frac{\frac{1}{N}\mathbf{K}_1 \circ \ldots \circ \frac{1}{N}\mathbf{K}_C}{\operatorname{tr}(\frac{1}{N}\mathbf{K}_1 \circ \ldots \circ \frac{1}{N}\mathbf{K}_C)} \\
&= \frac{1}{N^C} \frac{\mathbf{K}_1 \circ \ldots \circ \mathbf{K}_C}{\frac{1}{N^C}\operatorname{tr}(\mathbf{K}_1 \circ \ldots \circ \mathbf{K}_C)} \\
&= \frac{1}{N}\mathbf{K}_1 \circ \ldots \circ \mathbf{K}_C,
\end{aligned}
$$

since $\operatorname{tr}(\mathbf{K}_1 \circ \ldots \circ \mathbf{K}_C) = N$. Thus, element $(i, j)$ is given by

$$
\left(\frac{\mathbf{A}_1 \circ \ldots \circ \mathbf{A}_C}{\operatorname{tr}(\mathbf{A}_1 \circ \ldots \circ \mathbf{A}_C)}\right)_{ij} = \frac{1}{N}\prod_{c=1}^{C}(\mathbf{K}_c)_{ij} = \frac{1}{N}e^{-\sum_{c=1}^{C}\frac{1}{\sigma_c^2}\|\mathbf{x}_i^{(c)}-\mathbf{x}_j^{(c)}\|^2}.
$$

If we let $\sigma = \sigma_1 = \sigma_2 = \ldots = \sigma_C$, this expression is reduced to

$$
\frac{1}{N}e^{-\frac{1}{\sigma^2}\sum_{c=1}^{C}\|\mathbf{x}_i^{(c)}-\mathbf{x}_j^{(c)}\|^2} = \frac{1}{N}e^{-\frac{1}{\sigma^2}\|\mathbf{x}_i-\mathbf{x}_j\|^2} = \frac{1}{N}\kappa_{\text{ten}}(\mathbb{X}_i, \mathbb{X}_j). \tag{14}
$$

Accordingly, $S_\alpha(\mathbf{A}_{\text{ten}}) = S_\alpha(\mathbf{A}_1, \ldots, \mathbf{A}_C)$ implying that the tensor method is equivalent to the multivariate matrix–based joint entropy when the width parameter is equal within a given layer, assuming an RBF kernel is used. However, the tensor-based approach eliminates the effect of numerical instabilities one encounters in layers with many filters, thereby enabling training of complex neural networks.

## E    Numerical Instability of Multivariate Approach and Structure Preserving Tensor Kernels

As explained in Section 3.2, the multivariate approach of Yu et al. (2019) (Equation 6) struggles when the number of channels in an image tensor becomes large, as a result of the Hadamard products in Equation 6. To illustrate this instability we have conducted a simple example. A subset of 50 samples is extracted from the MNIST dataset. Then, each image is duplicated (plus some noise) $C$ times along the channel dimension of the same image, i.e. going from a grayscale image of size $(1, 1, 28, 28)$ to a new image of size $(1, C, 28, 28)$. Since the same image is added along the channel dimension the kernel matrix should not change dramatically. Figure 7 displays the results of the experiment just described. The first row if Figure 7 shows the kernel matrices based on the multivariate approach proposed by Yu et al. (2019). When the tensor data only has one channel (first column) the kernel obtained is identical to the results obtained using the tensor kernel described in this paper. However, as the number of channels increase, the off-diagonal quickly vanishes and the kernel matrix tends towards a diagonal matrix. This is a result of vanishing Hadamard products, as described in Section 3.2 of the main paper. Theoretically, the multivariate approach should yield the same kernel as with the tensor kernel approach, as explained in Section 4, but the off-diagonal elements decrease so quickly that they fall outside numerical precision. The second row of Figure 7 depicts the kernel matrices obtained using the tensor kernel approach described in Section 4. The kernel matrices in this row are almost unchanged as the number of channels increase, which is to be expected. Since the same image is added along the channel dimension, the similarity between the sample should not change drastically, which is what this row demonstrates. The third row of Figure 7 displays the kernel matrices obtained using so-called matricization-based tensor kernels Signoretto et al. (2011), which are tensor kernels that preserve structure between the channels of the tensor. In this case, this approach produces similar results to the tensor kernel used in this paper, which is to be expected. Since the same image is added along the channel dimension there is little information

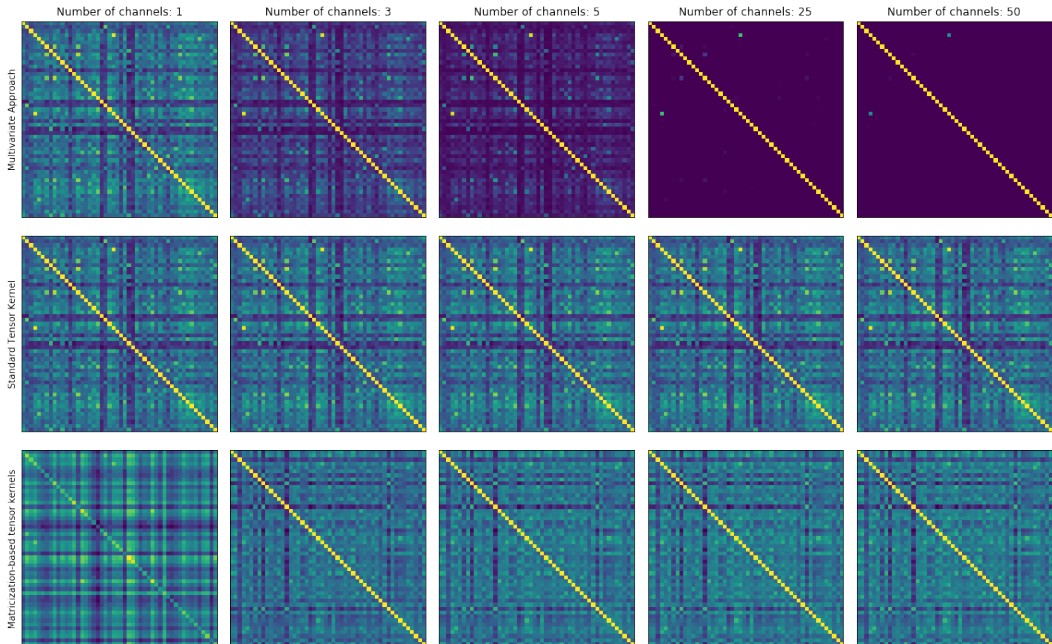

Figure 7: Different approaches for calculating kernels based on tensor data. First row shows the multivariate approach of Yu et al. (2019), second row depicts the tensor kernel approach used in this paper, and third row displays the kernel obtained using matricization-based tensor kernels Signoretto et al. (2011) that preserve structure between channels. Bright colors indicate high values while dark values indicate low values in all the kernel matrices.

to extract between the channels. We hypothesize that for small images with centered objects, such as with MNIST and CIFAR10, the structured tensor kernel does not capture much more information than the tensor kernel described in Section 4. However, for more complex tensor data, exploring the potential of such structure preserving tensor kernels is an interesting avenue for future studies.

## F    DETAILED DESCRIPTION OF NETWORKS FROM SECTION 5

We provide a detailed description of the architectures utilized in Section 5 of the main paper. Weights were initialized according to He et al. (2015) when the ReLU activation function was applied and initialized according to Glorot & Bengio (2010) for the experiments conducted using the tanh activation function. Biases were initialized as zeros for all networks. All networks were implemented using the deep learning framework Pytorch (Paszke et al., 2017).

### F.1    MULTILAYER PERCEPTRON UTILIZED IN SECTION 5

The MLP architecture used in our experiments is the same architecture utilized in previous studies on the IP of DNN (Noshad et al., 2019; Saxe et al., 2018), but with Batch Normalization (Ioffe & Szegedy, 2015) included after the activation function of each hidden layer. Specifically, the MLP in Section 5 includes (from input to output) the following components:

1. Fully connected layer with 784 inputs and 1024 outputs.
2. Activation function.
3. Batch normalization layer
4. Fully connected layer with 1024 inputs and 20 outputs.
5. Activation function.
6. Batch normalization layer
7. Fully connected layer with 20 inputs and 20 outputs.

8. Activation function.

9. Batch normalization layer

10. Fully connected layer with 20 inputs and 20 outputs.

11. Activation function.

12. Batch normalization layer

13. Fully connected layer with 784 inputs and 10 outputs.

14. Softmax activation function.

## F.2 CONVOLUTIONAL NEURAL NETWORK UTILIZED IN SECTION 5

The CNN architecture in our experiments is a similar architecture as the one used by Noshad et al. (2019). Specifically, the CNN in Section 5 includes (from input to output) the following components:

1. Convolutional layer with 1 input channel and 4 filters, filter size $3 \times 3$, stride of 1 and no padding.

2. Activation function.

3. Batch normalization layer

4. Convolutional layer with 4 input channels and 8 filters, filter size $3 \times 3$, stride of 1 and no padding.

5. Activation function.

6. Batch normalization layer

7. Max pooling layer with filter size $2 \times 2$, stride of 2 and no padding.

8. Convolutional layer with 8 input channels and 16 filters, filter size $3 \times 3$, stride of 1 and no padding.

9. Activation function.

10. Batch normalization layer

11. Max pooling layer with filter size $2 \times 2$, stride of 2 and no padding.

12. Fully connected layer with 400 inputs and 256 outputs.

13. Activation function.

14. Batch normalization layer

15. Fully connected layer with 256 inputs and 10 outputs.

16. Softmax activation function.

## G   IP OF MLP WITH TANH ACTIVATION FUNCTION FROM SECTION 5

Figure 8 displays the IP of the MLP described above with a tanh activation function applied in each hidden layer. Similarly to the ReLU experiment in the main paper, a fitting phase is observed, where both $I(T; X)$ and $I(Y; T)$ increases rapidly, followed by a compression phase where $I(T; X)$ decrease and $I(Y; T)$ remains unchanged. Also note that, similar to the ReLU experiment, $I(Y; T)$ stabilizes close to the theoretical maximum value of $\log_2(10)$.

## H   IP OF CNN WITH TANH ACTIVATION FUNCTION FROM SECTION 5

Figure 9 displays the IP of the CNN described above with a tanh activation function applied in each hidden layer. Just as for the CNN experiment with ReLU activation function in the main paper, no fitting phase is observed for the majority of the layers, which might indicate that the convolutional layers can extract the essential information after only a few iterations.

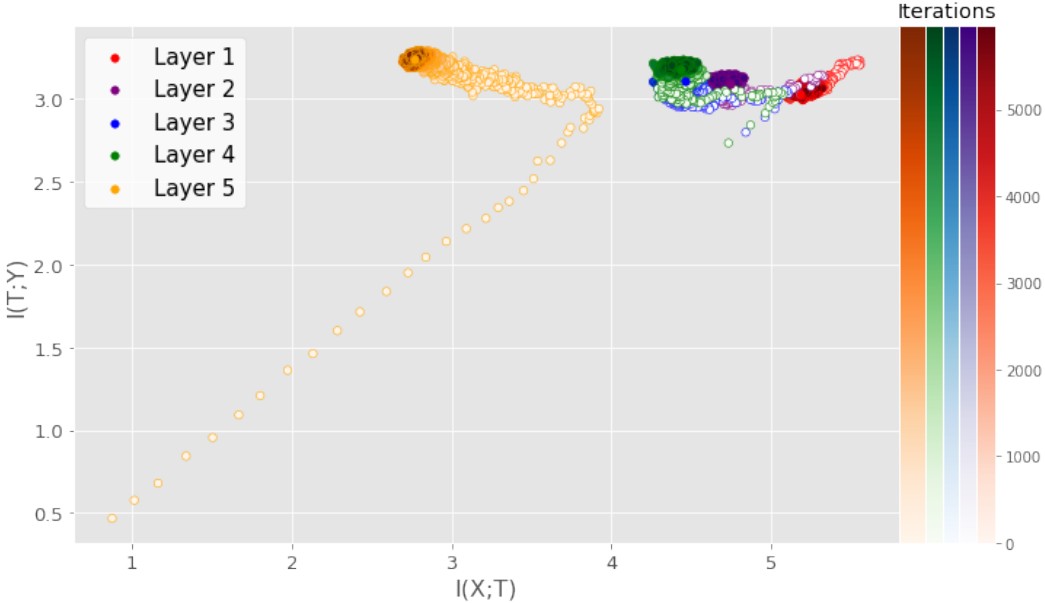

Figure 8: IP of a MLP consisting of four fully connected layers with 1024, 20, 20, and 20 neurons and a tanh activation function in in each hidden layer. MI was estimated using the training data of the MNIST dataset and averaged over 5 runs.

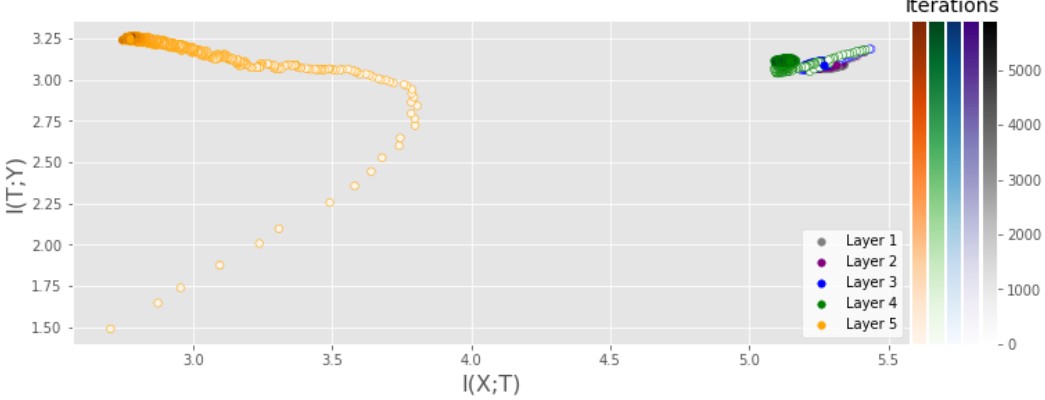

Figure 9: IP of a CNN consisting of three convolutional layers with 4, 8 and 12 filters and one fully connected layer with 256 neurons and a tanh activation function in in each hidden layer. MI was estimated using the training data of the MNIST dataset and averaged over 5 runs.

## I    KERNEL WIDTH SIGMA

We further evaluate our dynamic approach of finding the kernel width $\sigma$. Figure 10 shows the variation of $\sigma$ in each layer for the MLP, the small CNN and the VGG16 network. We observe that the optimal kernel width for each layer (based on the criterion in Section 4.2), stabilizes reasonably quickly and remains relatively constant during training. This illustrates that decreasing the sampling range is a meaningful approach to decrease computational complexity.

## J    INFORMATION PLANE VIDEO

To further investigate the role of the compression phase in the IP we have created a video that is available at the following link: `https://streamable.com/7gsxe`. The video shows the training process of a simple 3 layer perceptron with 2 ReLU neurons in each layer, which is trained

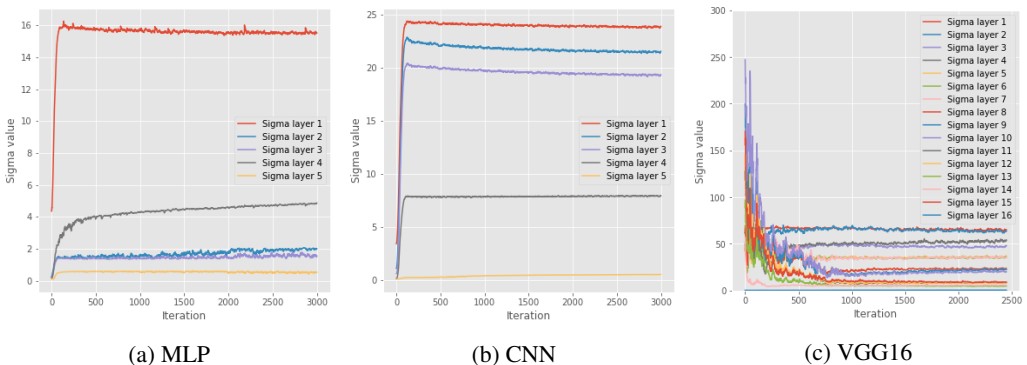

|  |  |  |
|---|---|---|
| (a) MLP | (b) CNN | (c) VGG16 |

Figure 10: Evolution of kernel width as a function of iteration for the three networks that we considered in this work. The plots demonstrate how the optimal kernel width quickly stabilizes and stay relatively stable throughout the training.

to solve the XOR problem. We used such a simple network as it allows us to visualize the decision surface of each layer and how the data is transformed throughout the training, as well as the information plane. All information theoretic quantities are calculated using the matrix based approach described in this paper.

Several interesting aspects could be highlighted, but we would like to focus on the compression phase. After approximately 1250 iteration the network approaches 100 percent accuracy. At the same time, the output layer starts to show compression. By inspecting the decision surface in the input space (plot shown in first row and first column of the video) it is clear that the network has found a solution that separates the two classes. Early stopping would stop the training at approximately 1300 iterations, depending on the patience parameter. At the end of the training, the decision surface in the input space has become much more narrow, thus leaving less opportunity to correctly classify new samples that lay slightly outside the space of the training samples. Such observations corroborate the suggestions from the main paper, that the compression phase is not necessarily associated with improved generalization.

## K    DATA PROCESSING INEQUALITY

Figure 11 illustrates the mean difference in MI between two subsequent layers in the MLP and VGG16 network. Positive numbers indicate that MI decreases, thus indicating compliance with the DPI. We observe that our estimator complies with the DPI for all layers in the MLP and for all except one in the VGG16 network.

Note, the difference in MI is considerably lower for the early layers in the network, which is further shown by the grouping of the early layers for our convolutional based architectures (Figure 2-4).

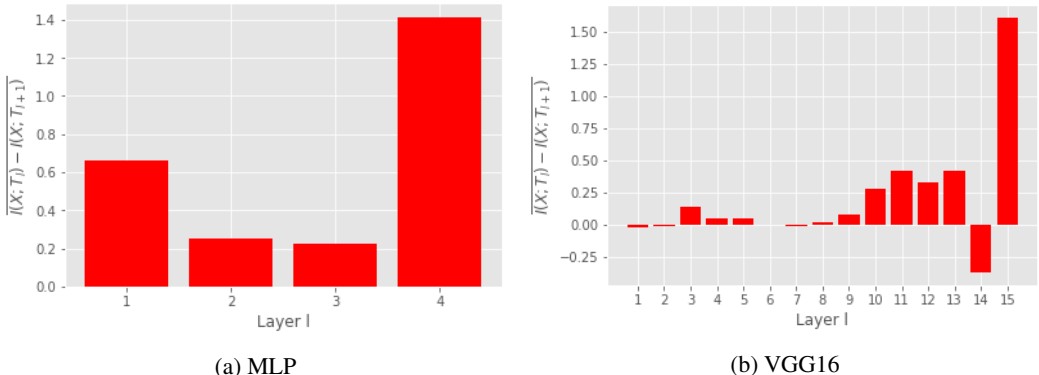

(a) MLP

(b) VGG16

Figure 11: Mean difference in MI of subsequent layers $\ell$ and $\ell + 1$. Positive numbers indicate compliance with the DPI. MI was estimated on the MNIST training set for the MLP and on the CIFAR-10 training set for the VGG16.

