# OpenReview forum: "Information Plane Analysis of Deep Neural Networks via Matrix--Based Renyi's Entropy and Tensor Kernels"
_ICLR.cc/2020/Conference — Reject_

### Official Review · AnonReviewer1 · 2019-10-23
**Official Blind Review #1**

**Rating:** 3

**Review:**

In this paper, the authors try to resolve the problem of estimating mutual information between high-dimensional layers in neural network. Specifically, the authors try to tackle the convolutional neural networks, by propose a novel tensor-kernel based estimator. They use the proposed estimator to discover the similar trends on the information plane as the previous works.

In general, I think the introduction of the tensor kernels for mutual information estimation is the key contribution of this paper. However, I think this contribution is a little bit incremental, compared to the multivariate matrix-based version introduced by Yu et.al. In formula (7), the authors use the Frobenius distance of two tensors as the input of the kernal, which is equivalent to vectorize the tensor and use Euclidean distance. This approach does not capture the special structure of tensor and seems incremental. Also, the phenomenon of "turning point" on information plane for neural network has been challenged since its first publication. Despite its high citation, the deep meaning of this phenomenon has not been clearly studied. So an incremental change on computing such a questionable phenomenon make the contribution of this paper not very strong.

So I would like to give a weak reject.

**Experience Assessment:**

I have published one or two papers in this area.

**Review Assessment: Checking Correctness Of Derivations And Theory:**

I assessed the sensibility of the derivations and theory.

**Review Assessment: Checking Correctness Of Experiments:**

I assessed the sensibility of the experiments.

**Review Assessment: Thoroughness In Paper Reading:**

I read the paper at least twice and used my best judgement in assessing the paper.

---

> ### Author Response · Authors · 2019-11-11
> **Answer to Reviewer 1**
>
> We would like to thank the reviewer for the valuable feedback on our paper. We have conducted experiments and revised the paper to address the reviewer's comments.
>
> - Comment 1: Novelty of tensor kernel.
>
> - Answer 1:
>
> We think it is important to emphasize that the multivariate approach of Yu et.al. [1] has a very different starting point than our approach. The fundamental idea of our proposed framework is not limited to the tensor kernel described in this paper and is numerically stable for large-scale neural networks where Yu et.al. fails [2]. More specifically, they consider each feature map produced by a filter in a convolutional layer as its own random variable, and then use multivariate information theory to estimate the shared information between these variables. Instead, our starting point is that the entire output of a convolutional layer is its own random variable, thus requiring a very different approach.
>
> Furthermore, it is certainly true that the chosen tensor kernel corresponds to simply vectorizing tensors and using a RBF kernel. However, it was chosen for its simplicity and computational benefits even though such an approach ignores inter-component structures within and between the respective tensors [3]. We have updated the paper to better motivate this choice and have further included a simple experiment in Appendix E of the supplementary material to illustrate that structure preserving tensor kernels can be used in a simple setting, limited only by computational complexity.
>
> - Comment 2: "Turning point" phenomenon.
>
> - Answer 2:
>
> We would like to highlight that our experiments on the "turning point" of the information plane of neural networks is only one part of our contributions. We examine the DPI of our proposed estimator for several networks and examine the claim of Cheng et.al. that $I(T;X)\approx H(X)$ [4]. Also, we propose novel methods for determining the bandwidth of the kernels used in our estimators, which we believe makes our estimators more stable and robust. However, we certainly agree that the details around this "turning point" have been highly debated, and are still not fully understood. Nevertheless, after the initial critique by Saxe et.al. [5], several studies [6, 7, 8] have reported similar results as Shwartz-Ziv and Thisby [9]. Seeing as most of these these studies have been focused on small networks on simple datasets, we investigate if a similar "turning point" would be observed in large networks on datasets were you do not achieve $\approx 100 \%$ accuracy. Furthermore, Shwartz-Ziv and Thisby argued that during the compression phase the network produces a representation with better generalization capabilities [9]. Our experiments suggest a link between the turning point and overfitting. But more research is required to fully understand the behavior of the IP.
>
> To further investigate the effect of early stopping and the compression phase of the IP, we have created a video of the evolution of the IP of a neural network during training. The video can be viewed by clicking the following link: https://streamable.com/7gsxe, and in Appendix J of the supplementary material we describe the experiment and discuss the findings which corroborate our initial hypothesis.
>
> [1] S. Yu, L. G. Sanchez Giraldo, R. Jenssen, and J. C. Principe. Multivariate extension of matrix-based renyi’s -order entropy functional. IEEE T-PAMI 2019.
>
> [2] S. Yu, K. Wickstrøm, R. Jenssen, and J. C. Principe.  Under-standing convolutional neural network training with information theory (2018).
>
> [3] M. Signoretto, L. De Lathauwer, and J.AK Suykens. A kernel-based framework to tensorial data analysis. Neural networks (2011).
>
> [4] H. Cheng, D. Lian, S. Gao, and Y. Geng. Evaluating capability of deep neural networks for image classification via information plane.  In ECCV September 2018.
>
> [5] A. Michael  Saxe,  Y.  Bansal,  J.  Dapello,  M.  Advani,  A. Kolchinsky, B. Daniel Tracey, and D. Daniel Cox.  On the information bottleneck  theory  of  deep  learning.   In ICLR 2018.
>
> [6] I. Chelombiev,  C. Houghton,  and Cian O’Donnell.   Adaptive estimators show information compression in deep neural networks. In ICLR, 2019.
>
> [7] M. Noshad, Y. Zeng, and A. O. Hero.   Scalable mutual information estimation using dependence graphs. In ICASSP 2019 - 2019 IEEE ICASSP.
>
> [8] S. Yu and J. C. Principe.  Understanding autoencoders with information theoretic concepts. Neural Networks.
>
> [9] R. Shwartz-Ziv and Naftali Tishby.   Opening the black box of deep neuralnetworks via information.

---

### Official Review · AnonReviewer2 · 2019-10-23
**Official Blind Review #2**

**Rating:** 6

**Review:**

=== SUMMARY ===

The paper...
- suggests an estimation method of mutual information that is less susceptible to numerical problems when many neurons are involved
- suggests an estimation method tailored to convolutional neural networks.
- analyses whether H(X) ≈ I(T;X) resp. H(Y) ≈ I(T;Y) is true in high dimensions for their suggested estimator.
- analyses how common regularization techniques affect the learning dynamics. They show that early stopping prevents the learning process from entering the compression phase


=== RELATED WORK ===

Prior work is clearly presented and well-treated. The introduction and related work give a good overview of prior work on information plane theory, touching on the important contributions from Tishby & Zaslavsky [2015], Schwartz Ziv & Tishby [2017], Saxe et. al [2018], and Noshad et. al [2019], among others. In Section 3, Renyi’s \alpha-order entropy is outlined, and the multi-variate extension proposed by Yu et al. [2019] is described.


=== APPROACH ===
Multiple papers are put into perspective, and the findings of these papers serve as a foundation for the suggested method
- Renyi’s Entropy is presented as a generalization of different entropy measures (among them Shannon Entropy). Computing Renyi’s Entropy however necessitates to get a high-dimensional density estimation over the data distribution
- Giraldo et. al. suggested a kernel-based method to estimate Renyi’s Entropy solely on data without estimating the (intractable) density. An estimation for the mutual information can also be found in said paper
- Yu et al. suggested a generalization of b) that can handle C variables instead of only two. As this computation includes taking the product over C values that lie within (0,1/N) the result tends to 0.

An RBF-based kernel is defined for a fixed conv. layer by computing the Frobenius norm w.r.t two 3D-output tensors of the conv. layer X_i, X_j. A derived matrix A is defined similarly to Giraldo et al.

This way, C individual kernels are defined, which are combined according to Yu et al.


=== EXPERIMENTS ===

Comparison to Previous Approaches
- Very detailed and clear structure of experiment.

Increasing DNN size
- The experiment shows that there is both a fitting and a compression phase
- H(X) ≈ I(T;X) resp. H(Y) ≈ I(T;Y) was disproven

Effect of Early Stopping
- Early Stopping prevents the learning process from entering the compression phase. The authors conjecture that the compression phase is related to overfitting -- it would be interesting to see some evidence to support this.

Data Processing Inequality
- Enough evidence is provided to sustain that the data processing equality also holds for the suggested estimator


=== OTHER COMMENTS ===

The paper would benefit from providing more rationale on why the suggested estimation method is numerically stable (as opposed to Yu et al.), maybe by supplying another proof.

Technical mistakes
Section 4.1. Introduces X_i as a 3D tensor of dimensions C x H x W. Eq (7) mentions the Frobenius norm which is defined over matrices. We assume that X_i should be rather defined as a 2D tensor of dimensions H x W for a fixed layer index after reading Section 4.2.


**Experience Assessment:**

I have read many papers in this area.

**Review Assessment: Checking Correctness Of Derivations And Theory:**

I assessed the sensibility of the derivations and theory.

**Review Assessment: Checking Correctness Of Experiments:**

I assessed the sensibility of the experiments.

**Review Assessment: Thoroughness In Paper Reading:**

I read the paper at least twice and used my best judgement in assessing the paper.

---

> ### Author Response · Authors · 2019-11-11
> **Answer to Reviewer 2**
>
> We appreciate the helpful comments provided by the reviewer. We have conducted experiments to address the reviewers' comments, which we have included in the revised manuscript.
>
> - Comment 1: Rationale on numerical stability of multivariate approach.
>
> - Answer 1:
>
> In order to illustrate the instability of the multivariate approach proposed by Yu et.al. [1], we conducted a simple experiment to demonstrate the advantage of our tensor based approach. This experiment has been included in Appendix E of the supplementary material of the revised manuscript where we describe the experiment and discuss the findings that demonstrate the superiority of the proposed approach.
>
> - Comment 2: Further investigation of connection between early-stopping, overfitting and compression phase.
>
> - Answer 2:
>
> To further investigate the effect of early stopping and the compression phase of the IP, we have created a video of the evolution of the IP during training. The video can be viewed by clicking the following link: https://streamable.com/7gsxe, and in Appendix J of the supplementary material we describe the experiment and discuss the findings which corroborate our initial hypothesis.
>
> - Comment 3: Technical mistake.
>
> - Answer 3:
>
> You are correct, the Frobenius norm in Eq (7) is actually the Hilbert-Frobenius norm, and is defined for rank 2 tensors. We have corrected this in the revised manuscript.
>
> [1] S. Yu, L. G. Sanchez Giraldo, R. Jenssen, and J. C. Principe. Multivariate extension of matrix-based renyi’s -order entropy functional. IEEE Transactions on Pattern Analysis and Machine Intelligence, 2019.

---

### Official Review · AnonReviewer3 · 2019-11-04
**Official Blind Review #3**

**Rating:** 6

**Review:**

This paper concerns the "information plane" view of visualizing and understanding neural net training. Kernel-based estimators of Renyi entropy-like quantities are applied to study moderate-sized DNNs. A straightforward extension to CNNs is presented.

Major comments:

I found this paper extremely hard to follow. I think a lot of my difficulty was a lack of familiarity with the papers this work was building on, but I also felt the main line of reasoning was unnecessarily opaque. I've tried to indicate where some clarifying wording might help readers such as myself.

In general I am very skeptical that reasonable entropy-like quantities can be estimated reliably for high-dimensional data using anything along the lines of kernel density estimation, especially based on a single minibatch or small collection of minibatches! The authors provide no experimental evidence that these estimates are even close to being accurate (for example on synthetic datasets where the true entropy / mutual information is known). Clearly the estimated quantities evolve during training, and that may be interesting in itself, but calling the estimated quantities "mutual information" seems like a leap that's not justified in the paper.

Throughout the paper, CNNs are referred to as having some special difficulty for entropy estimates because of the channel dimension. This is misleading. For DNNs the activations are vectors, and so a kernel defined over rank 1 tensors is needed. Even without the channel dimension, CNN activations would be rank 2 tensors, and so a kernel would need to be defined over rank 2 tensors. Going from rank 2 to rank 3 doesn't pose any special difficulties. Indeed the most obvious way of defining a kernel over higher rank tensors is to flatten them and use a kernel defined over vectors, which is exactly what the paper does in practice.


Minor comments:

In the introduction, it would be helpful to include some relevant citations after the sentence "Several works have demonstrated that one may unveil interesting properties... IP".

In section 3, the multivariate extension proposed by Yu et al. (2019) seems like an interesting side note (since it was used in a previous attempt to estimate information plane for CNNs), but it doesn't seem central to the paper, and I personally found it unnecessarily confusing to have it presented in the main text. What about moving sections 3.2 and 4.2 to the appendix for clarity?

In section 3.1, "over a finite set" is probably incorrect ("probability density function" implies a continuous space for X, as does the integral in (1)).

In section 3.1 (and the appendix), "The ones developed for Shannon" seems imprecise. "Certain approaches developed for estimating the Shannon entropy"?

It's not clear what "have the same functional form of the statistical quantity" means. Which statistical quantity? What aspects of the functional form are similar? Please elaborate in the paper.

I think "marginal distribution" is incorrect. It's representing a whole probability distribution, not just a marginal distribution. Which marginal would it be in any case?

Section 3.2 states "The matrix-based... entropy functional is not suitable for estimating... convolutional layer... as the output consists of C feature maps", but as discussed above, there is no special difficulty caused by the channel dimension. Even if the channel dimension were not present the difficulties would be the same. (Also, it seems like defining a kernel over the rank-3 tensors is an extremely natural / unsurprising thing to try given the set-up so far.)

In section 4.1, "can not include tensor data without modifications" seems misleading for a similar reason. One of the great things about kernels is that they can be defined on lots of objects, including things like rank 3 tensors!

Near (8), it would be very helpful to state explicitly what is done for the joint entropy term in (5). It sounds like this term, which involves the Hadamard product, in practice amounts to summing the Euclidean distances between x's and between y's, and it might be helpful to the new reader to point this out. (It also highlights that the method is easy to implement in practice).

The discussion in section 4.2 is only valid for the RBF kernel, but the first paragraph of that section makes it sound like it is true more generally.

At the bottom of section 4.2, if the proposed approach is equivalent to the multivariate approach, then how can one suffer from numerical instability while the other doesn't? Also, numerical stability makes it sound like an artifact of floating point arithmetic, whereas I think the point that's being made is more mathematical? Please clarify in the paper.

In "enabling training of complex neural networks", shouldn't "training" be "analysis"?

In section 5, under "increasing DNN size", I wasn't clear on the meaning of "deterministic" in "neither .... is deterministic for our proposed estimator". Random variables can be deterministic or not, but how can a mutual information be deterministic?

Under "effect of early stopping", isn't looking at the test set entropies (as is done elsewhere in the paper) much more relevant to overfitting than considering different "patience" values?

In the bibliography, two different papers are "Yu et al. (2019)".

In appendix A, "the same functional form of the statistical quantity", "marginal", etc don't seem quite correct, as mentioned above. Also the first equation should not have a comma between f(X) and g(Y) (which if I understand correctly are being multiplied).

**Experience Assessment:**

I do not know much about this area.

**Review Assessment: Checking Correctness Of Derivations And Theory:**

I assessed the sensibility of the derivations and theory.

**Review Assessment: Checking Correctness Of Experiments:**

I assessed the sensibility of the experiments.

**Review Assessment: Thoroughness In Paper Reading:**

I read the paper thoroughly.

---

> ### Author Response · Authors · 2019-11-11
> **Answers to Reviewer 3 (Part 2)**
>
> - Comment 5: Imprecise formulation ("have the same functional form of the statistical quantity"), "marginal", comma between f(X) and g(Y).
>
> - Answer 5: We have taken the liberty to collect three of your comments so they can be answered simultaneously. First, we acknowledge that the sentence "have the same functional form of the statistical quantity” could have been more transparent. We have removed it and rewritten the last part of Section 3.1 and Appendix B to make the text more clear. Further, we have correct the use of the term "marginal" and the comma typo in the revised manuscript. Your understanding is correct and f(X) and g(y) are multiplied in the equation.
>
> - Comment 6: Explicit formulation of joint entropy.
>
> - Answer 6: We have modified Section 4.1 to explicitly state how the entropy and joint entropy is computed.
>
> - Comment 7: Imprecise formulation (Discussion in section 4.2  only valid for the RBF kernel).
>
> - Answer 7: The discussion is indeed only valid for the RBF kernel, which should have been clarified at the beginning of the first paragraph. We have clarified this in the revised manuscript.
>
> - Comment 8: Numerical instability of multivariate approach.
>
> - Answer 8: As pointed out in our previous comment, one important difference between Yu et.al. [1] and our approach is the starting point. This difference in starting point leads to a difference when the kernel matrix is computed. Although both should end up with identical results, the difference in computation leads to issues as the number of channels increases. More precisely, as the number of channels increases off-diagonal elements in the Hadamard product of kernel matrices approach zero. Although the normalization coefficient should correct this, in practice, this does not happen due to numerical precision and results in a diagonal kernel matrix. In order to illustrate the instability of the multivariate approach proposed by Yu et.al. [1], we conducted a simple experiment to demonstrate the advantage of our tensor based approach. This experiment has been included in Appendix E of the supplementary material of the revised manuscript where we describe the experiment.
>
> - Comment 9: "training" -> "analysis"?
>
> - Answer 9: You are correct, it should be "analysis". We have corrected this in the revised manuscript.
>
> - Comment 10: Meaning of deterministic.
>
> - Answer 10: Deterministic was used to match the terminology of Cheng et al. [2], who use it to refer to the claim that $I(X;T)\approx H(X)$ and $I(T;Y)\approx H(Y)$ for high dimensional data and in a larger picture is based on the deterministic information bottleneck [3]. Essentially, they state that the mutual information between the input/labels and the hidden representation does not change at all during training and will just remain constant. We show that this does not hold for our estimator.
>
> - Comment 11:Test set entropies for early stopping experiment.
>
> - Answer 11: Following the convention of previous works [4] and to be able to compare against them, we have chosen to visualize the IP for training data. It is reasonable to examine the training data as this is the data it would overfit on. However, we did perform the same experiment on the test data and got similar results. The figure shown by clicking the following link https://pasteboard.co/IGaVrFh.png displays the corresponding plot to Figure 1 in the main paper but with all quantities estimated on the test data. While the stopping points are not marked in this plot, they coincide closely with the stopping points labeled in Figure 1 of the main paper.
> Note, as requested by Reviewer 1 and 2, we added an experiment to further investigate the effect of early stopping and the compression phase of the IP. We have created a video of the evolution of the IP of a neural network during training. The video can be viewed by clicking the following link: https://streamable.com/7gsxe, and in Appendix J of the supplementary material we describe the experiment and discuss the findings which corroborate our initial hypothesis.
>
> - Comment 12: Bibliography error.
>
> - Answer 12: Only one paper should be denoted as Yu et al. (2019), namely the "Multivariate extension of matrixbased renyi’s alpha-order entropy functional". The second paper, "Understanding autoencoders with information theory", only has two authors and is therefore denoted as Yu and Principe (2019).
>
> [1] S. Yu, L. G. Sanchez Giraldo, R. Jenssen, and J. C. Principe. Multivariate ex-tension of matrix-based renyi’s -order entropy functional.IEEE Transactions onPattern Analysis and Machine Intelligence, pages 1–1, 2019.
>
> [2] H. Cheng, D. Lian, S. Gao, and Y. Geng. Evaluating capability of deep neural networks for image classification via information plane.  ECCV September 2018.
>
> [3] D J Strouse and Dd J Schwab.  The deterministic information bottleneck. UAI 2016.
>
> [4] M. Noshad, Y. Zeng, and A. O. Hero.   Scalable mutual information estimation using dependence graphs. ICASSP 2019

---

> ### Author Response · Authors · 2019-11-11
> **Answer to Reviewer 3 (Part 1)**
>
> First of all, we would like to thank the reviewer for the time and effort gone into producing such a thorough review. We have provided answers to the reviewers' comments and conducted experiments to try and shed light on unclear aspects of the paper. We have incorporated your suggestions and suggestions from the other two reviewers in the revised manuscript, which we hope has made the text more transparent.
>
> Major Comments:
>
> - Comment 1: Estimating information theoretic quantities of high-dimensional data.
>
> - Answer 1:
>
> There is no doubt that estimating information theoretic quantities of high-dimensional data is a challenging task. We acknowledge that our paper could have benefited from experiments on synthetic data. In Appendix A of the supplementary material we now report the results of two simple experiments that demonstrate how the estimators used in this paper behave in high-dimensional space and in the mini-batch setting.
>
> Furthermore, we would like to emphasize that, while obtaining an exact estimate for information theoretic quantities in high-dimensional space is very challenging, examining the general patterns of the quantities might be more attainable. Also, we should highlight that we do not use the kernel as a density estimator. Instead, we build a density matrix, similar to that of quantum information theory [1], from raw data and measure the desired quantity in a RKHS. We have modified the last part of Section 3.1 to better reflect these aspects.
>
> - Comment 2: Novelty of tensor kernel / vectorization of tensors.
>
> - Answer 2:
>
> We would like to point out that the starting points of our approach and Yu et.al. (2019) [2] are inherently different as we model each layer as a single random variable, while they model each layer as a set of channels, where each channel is represented by a random variable. This particular difference in approach allows our proposed method to be scalable for an increasing number of channels (see Appendix E).
>
> Note, Yu et.al. (2018) [3] vectorize each individual channel, thereby reducing the rank 2 tensors to rank 1 tensors and thereby distorts spatial relationships [3]. This leads to the problem with the channel dimensions as rank 3 tensors are now reduced to rank 2 tensors in their framework. Our approach, instead inherently incorporates the possibility for modelling inter-component structures such as spatial relationships and channel relationships in the choice of tensor metric.
>
> In this work, we apply the tensor kernel described in Section 4.1 due to its simplicity and computational efficiency, which makes the approach easy to implement and thereby more usable. The differences between the two approaches, for the chosen kernel, result in simply vectorizing tensors and using a RBF kernel in our method. However, the framework in itself generalizes to more complex tensor measures that for instance are able to exploit inter-component structures within and between the respective tensor [4]. We have added an experiment in Appendix E to highlight the possibilities that our tensor framework presents and motivated our choice of kernel in a better way.
>
> ------------------------------------------------------
>
> Minor Comments:
>
> - Comment 1: Relevant citations in introduction.
>
> - Answer 1: We have added several relevant citations in the revised manuscript.
>
> - Comment 2: Moving Section 3.2 and 4.2 to supplementary for clarity.
>
> - Answer 2: In order to avoid the confusion, we have moved section 4.2 to the supplementary material (Appendix D). We agree that it is an interesting side note, but not necessarily crucial to the main paper. However, we believe the multivariate approach described in Section 3.2 makes the reader more familiar with the general family of matrix-based functionals and therefore decided to keep the section in main body of the paper.
>
> - Comment 3: In section 3.1: "over a finite set" is probably incorrect.
>
> - Answer 3: You are correct and we have corrected the formulation in the revised manuscript.
>
> - Comment 4: Imprecise formulation ("The ones developed for Shannon" seems imprecise).
>
> - Answer 4: We agree that this formulation was not as clear as it should have been. Thank you for your suggestions, we have incorporated it in the revised manuscript.
>
> [1] M. A. Nielsen and I. L. Chuang. Quantum Computation and Quantum Information: 10th Anniversary Edition.  2011.
>
> [2] S. Yu, L. G. Sanchez Giraldo, R. Jenssen, and J. C. Principe. Multivariate extension of matrix-based Renyi’s -order entropy functional. IEEE TT-PAMI, 2019.
>
> [3] S. Yu, K. Wickstrøm, R. Jenssen, and J.C. Principe.  Understanding convolutional neural network training with information theory. 2018.
>
> [4] M. Signoretto, L. De Lathauwer, and J. AK Suykens. A kernel-based framework to tensorial data analysis.Neural networks, 2011.

---

### Decision · Program_Chairs · 2019-12-19

**Decision:**

Reject

**Comment:**

This paper considers the information plane analysis of DNNs. Estimating mutual information is required in such analysis which is difficult task for high dimensional problems. This paper proposes a new "matrix–based Renyi’s entropy coupled with ´tensor kernels over convolutional layers" to solve this problem. The methods seems to be related to an existing approach but derived using a different "starting point". Overall, the method is able to show improvements in high-dimensional case.

Both R1 and R3 have been critical of the approach. R3 is not convinced that the method would work for high-dimensional case and also that no simulation studies were provided. In the revised version the authors added a new experiment to show this. R3's another comment makes an interesting point regarding "the estimated quantities evolve during training, and that may be interesting in itself, but calling the estimated quantities mutual information seems like a leap that's not justified in the paper." I could not find an answer in the rebuttal regarding this.

R1 has also commented that the contribution is incremental in light of existing work. The authors mostly agree with this, but insist that the method is derived differently.

Overall, I think this is a reasonable paper with some minor issues. I think this can use another review cycle where the paper can be improved with additional results and to take care of some of the doubts that reviewers' had this time.

For now, I recommend to reject this paper, but encourage the authors to resubmit at another venue after revision.